# Using community health workers for facility and community based TB case finding: An evaluation in central Mozambique

B. José[1], I. Manhiça[1], J. Jones[1], C. Mutaquiha[1], P. Zindoga[1], I. Eduardo[2], J. Creswell[3], Z. Z. Qin[3], O. Ramis[3], I. Ramiro[4], M. Chidacua[4], J. Cowan[4,5]*

1 National TB Program, Mozambique Ministry of Health, Maputo, Mozambique, 2 Provincial TB Program, Mozambique Ministry of Health, Manica, Mozambique, 3 TB REACH, Stop TB Partnership, Geneva, Switzerland, 4 Health Alliance International, Beira, Mozambique, 5 Department of Global Health, University of Washington, Seattle, WA, United States of America

* jcowan22@gmail.com

**Data Availability Statement:** All relevant data are within the manuscript and its Supporting Information files. The TB Program Manager in Mozambique would like to be informed when any

## Abstract

### Background

Mozambique has one of the highest incidence rates of both TB and HIV in the world and an estimated tuberculosis (TB) treatment coverage of only 57% in 2018. Numerous approaches are being tested to reduce existing gaps in coverage and the estimated number of missing cases.

### Methods

Thirty Community Healthcare Workers (CHWs) were tasked with increasing TB notifications by performing verbal facility-based TB screening of all people presenting for care and TB contact tracing in the community. Using routine National TB Program data, we analyzed trends in TB notifications in five intervention districts and seven control districts in Manica province the year before this project and during a one-year intervention period.

### Results

In the four quarters before the study, the intervention districts notified 5,219 individuals with all forms of TB, and the control districts notified 2,248 TB cases. During the study 5,982 all forms of people with TB were notified in the intervention area, an increase of 763 (14.6%) over the baseline, whereas the control districts notified 1,877 persons with TB, a decrease of -371 (-16.5%). The CHW screening activities yielded 1,502 notified and treated individuals with TB.

### Conclusions

Employing CHWs to promote facility-based TB screening and household contact tracing may lead to an overall increase in TB notification.

non-Mozambique Ministry of Health organization is accessing and analyzing TB data from Mozambique. The readers request that any group accessing this data please, as a courtesy, inform Dr. Ivan Manhiça at ivanmca2004@yahoo.com.br if possible.

**Funding:** This study was funded by Global Affairs Canada, The Bill and Melinda Gates Foundation and USAID through Stop TB Partnership's TB REACH initiative. JCreswell and ZZQ received salary from Global Affairs Canada. This study was also supported in part by a 2015 developmental grant from the University of Washington Center for AIDS Research (CFAR), an NIH funded program under award number AI027757 which is supported by the following NIH Institutes and Centers (NIAID, NCI, NIMH, NIDA, NICHD, NHLBI, NIA, NIGMS, NIDDK). This funder provided salary support for JCowan. Funders did not have any additional role in the study design, data collection and analysis, decision to publish, or preparation of the manuscript. The specific roles of authors are articulated in the 'author contributions' section. There was no additional external funding received for this study. The content is solely the responsibility of the authors and does not necessarily represent the official views of the National Institutes of Health, Global Affairs Canada, USAID, The Bill and Melinda Gases Foundation, USAID or the Stop TB Partnership.

**Competing interests:** There are no competing Interests to declare.

# Introduction

Tuberculosis (TB) continues to be an epidemic in Sub-Saharan Africa and the leading infectious cause of death globally. Mozambique is one of the 30 high burden tuberculosis countries. The World Health Organization (WHO) estimates its TB incidence rate at 551/100,000 in 2018 translating to an estimate of 162,000 new individuals each year. [1] WHO estimates are based on its assessment of data notification quality and coverage, prevalence of the disease and information on death registration. In Mozambique 92,381 people were diagnosed and notified with TB in 2018, resulting in a treatment coverage rate of only 57%. Coverage for multi-drug resistant or rifampin resistant TB (MDR/RR-TB) was even lower, 14% (1,134 people notified with RR/MDR-TB of the estimated 8,300 total). [1] TB continues to be a leading cause of death in Mozambique, and the primary cause of death and disability among people living with HIV. [2–4] The high incidence rate was fueled by the HIV epidemic with a national prevalence of 13.2% and compounded by poverty. The estimated average gross national income per capita is only $440 USD, one of the lowest in Africa. [5–7]

TB case detection, diagnosis and notification is primarily conducted in health centers using passive case finding (waiting for the individual to present with TB-related symptoms) and testing with smear microscopy (SS) or increasingly using Xpert MTB/RIF (Xpert) testing as recommended by the WHO. [8] Following WHO guidelines, notification includes bacteriologically confirmed individuals and those clinically diagnosed in absence of positive tests or in absence of bacteriological test when it is unavailable. [9]

Multiple studies have documented significant individual patient and health system delays in TB diagnosis in Mozambique. [10, 11] People with TB often experience barriers to access the health facilities, first seek care from traditional healers, and/or visit health facilities several times before being tested and diagnosed with TB. This may partly explaining the gap between the actual number of people diagnosed and notified by the health system and the estimated number of people with TB in Mozambique. There is also a gap between the number of people diagnosed with and the number of people notified with TB–this is referred to as pre-treatment lost to follow-up (PTFLU). [10–14]

We hypothesized the following situations occurred in the districts where we worked: 1) a number of individuals with TB who present to health facilities are not diagnosed; 2) TB contact tracing could be optimized if Community Health Workers (CHWs) were given a role in it; 3) CHWs could identify individuals that are PTLTFU (people who are tested and have a positive laboratory test for TB, but are not subsequently informed of their results and do not begin treatment for TB) and 4) CHWs can help relink individuals that stop their anti-TB therapy or are lost to follow-up (LTFU) to care. We wanted to evaluate if interventions performed by CHWs to improve all these four weaknesses could lead to an increase in TB notifications at the district level.

Given the burden of TB and MDR/RR-TB, active case finding, improved diagnostics and access to treatment are a priority for the Mozambican National Tuberculosis Program (NTP) as part of an effort to reach the people with TB who are missed in Mozambique. As a result, Health Alliance International (HAI), a global health NGO with over 25 years of experience working in Mozambique, in partnership with the NTP, applied for and received funding for a TB REACH case finding project.

# Methodology

Through TB REACH funding, the Mozambican NTP and HAI employed thirty CHWs in five intervention districts of Manica Province (Gondola, Manica, Mossurize, Bárue, and Chimoio districts) where no other active case finding intervention was deployed. Candidates were

identified, screened based on basic literacy and ability to use a smartphone, interviewed, and the final CHWs selected in partnership with district and facility TB supervisors and community leaders. They underwent a weeklong training using a ministry of health approved training package for TB CHWs which included instructions and a practicum about how to lead short TB sensitization sessions for all individuals presenting to and waiting for care at 14 health facilities (including the five district hospital/health centers and nine peripheral health centers) in the districts. CHWs were trained to perform systematic five symptom screening (persistent cough, weight loss, night sweats, fever or hemoptysis as recommended by the WHO) of all people entering these health centers and for household members of TB contacts. [15] People providing positive answers to symptom screening were referred for sputum sample collection and testing. Samples were analyzed by Xpert or smear microscopy depending on the availability in each facility. In addition to facility-based TB screening, the CHWs were trained in and also responsible for active contact tracing of people diagnosed and notified with TB through household visits where they also provided short TB sensitization sessions and WHO recommended symptom screening. CHWs worked with health facilities to identify individuals known to be PTLTFU or LTFU and then doing community based tracing to find and link these individuals to care. CHWs were supervised by the facility TB nurse, district TB supervisor from the NTP, and the project field supervisor. CHWs completed summary reports of their monthly activities which were reviewed and validated by the facility TB nurse and project field supervisor before being digitized.

Using routine NTP data we analyzed trends in quarterly TB notifications in the five Manica Province intervention districts (Gondola, Manica, Mossurize, Bárue, Chimoio–total population 1.8 million) and in seven Manica Province control districts (Sussendenga, Machaze, Guro, Tambara, Macossa, Macate, Vanduze–total population 1.6 million) in the 12 quarters (Q4 2014 through Q3 2017) before this project and during the four quarter intervention period (Q4 2017 through Q3 2018). Intervention and control districts were purposely selected by the provincial health department in an effort to focus CHW activities in districts with larger populations that may not have effective TB community partners. Control districts were selected among non-intervention districts and were characterized for not receiving any active TB case finding activity during the study and being roughly similar demographically and in terms of health care development to the intervention districts.

We analyzed and cross-referenced NTP data using monthly reports summarizing the activities of each individual CHWs that were based on a daily register of recorded activities. This CHW register captured: the number of TB sensitization sessions, the number of participants in TB trainings, a list of individuals with presumptive TB identified by the CHW, a list of individuals noted to be PTLTFU and LTFU who the CHW was responsible for tracing.

## Analysis

Routine NTP notification data was analyzed using a pre-post evaluation methodology of total TB and bacteriologically confirmed (B+) notifications in intervention and control districts. The analysis was part of the routine project monitoring and evaluation framework, and followed the standard TB REACH methodology to determine the impact of active case finding in notification and the additional number of people with TB that were notified and that could be attributed to the specific intervention. [16] The direct yield (TB cases found by CHWs) of the intervention was tracked using the reports from the CHWs and we evaluated the impact of the direct yield on overall NTP reported TB notifications (including passive case finding). We compared the number of new B+ and all forms of people with TB notified during the baseline period (Q4 2016 thru Q3 2017) to the notifications during the intervention period (Q4 2017 to

Q3 2018) and then adjusted for historical TB notification trends during the previous three years (Q4 2014 thru Q3 2017) in both the intervention and the control districts. We estimated the expected notification during the implementation period in absence of intervention both in the districts where the project was implemented and in the control districts by applying linear regression to fit a trend line using the baseline notification data. Estimates were later compared with the actual notification values to measure the additional number of identified people notified with TB using a regression analysis and adjusting for historical trends. The relative changes in notifications in the intervention areas were further compared with the changes in notification in the control area between the same periods. This methodology is described in further detail elsewhere. [16, 17]

The intervention was approved by the National and Provincial TB Programs. Since the primary purpose of this evaluation was not research but to measure the increase the number of individuals notified with TB, and that the evaluation framework for this project and data analysis involved routine NTP data and project registries, formal review was not required by the University of Washington or Mozambican IRB.

## Results

During the four quarters of case-finding activities, the 30 CHWs led 6,737 TB sensitization sessions at facilities, in communities, schools and in TB contact households. This is approximately 19 sessions per CHW per month (or one per workday). CHWs reported that over 277,917 individuals participated in these sensitization sessions (some were at large public events attended by hundreds of people). As part of these sessions and active case finding activities CHWs identified 8,532 individuals with presumptive TB who they recorded in their registers, and successfully referred 7,921 (93%) for laboratory testing and/or clinical evaluation. Of these, 7,205 (84% of all people with presumptive TB) had a valid sputum smear microscopy or Xpert MTB/RIF test result. Eventually, 1,508 (18%) of all the presumptive individuals with TB were diagnosed with TB including 814 (54%) with bacteriological confirmation either by Xpert (577) or sputum smear microscopy (237). Almost all people diagnosed with TB as a result of this intervention started treatment (99.6%). The final yield of this intervention was 806 B+ individuals, and 1,502 people with all forms of TB diagnosed and enrolled in treatment.

The 30 CHWs also visited a total of 1,123 index case households, and 826 children less that 5 years old, who were contacts of individuals notified with TB and did symptom screening and referral for those who screened positive. Of the 826 children less than 5 years old, 703 were deemed eligible for and initiated TB preventive therapy with isoniazid. The number of people with active TB identified among household contacts and community case-finding activities was not recorded separately as part of this project, but routine NTP data shows that in the intervention districts as a whole less than 20% of individuals notified with TB were from contact tracing or community-based (not health facility-based) case finding activities.

During the study, the CHWs were also tasked with tracing 169 individuals who were pre-treatment lost to follow up (PTLTFU), of whom 148 (88%) were found, notified and started treatment for TB. The CHWs also identified 147 individuals who began but interrupted TB treatment, of whom 121 (85%) were found, relinked to care and restarted TB treatment. Process indicators for CHW TB activities are summarized in Table 1.

### Impact on notification

The year prior to the intervention 2,277 B+ and 5,219 all forms of TB were notified in the intervention area and 945 B+ and 2,248 all forms of TB in the control area. During this year-long study, the numbers of notifications increased to 2,933 B+ and 5,982 all forms of TB (an

**Table 1. TB Community Healthcare Worker (CHW) process indicators for thirty CHWs by quarter: Data reported by each CHW monthly based on their individual CHW TB activities register.**

| Indicator | Q4 2017 | Q1 2018 | Q2 2018 | Q3 2018 | Total |
|---|---|---|---|---|---|
| Total number of CHW trainings at health facilities and in the community | 1,073 | 1,725 | 1,984 | 1,955 | 6,737 |
| Total Number of participants in CHW led trainings | 40,531 | 76,871 | 79,821 | 80,694 | 277,917 |
| Total Number of Presumptive TB Cases (Positive Symptom Screen) Identified by CHWs | 1,267 | 2,195 | 2,653 | 2,417 | 8,532 |
| Total number of presumptive TB cases referred for laboratory testing and clinical evaluation | 1,067 | 2,106 | 2,494 | 2,254 | 7,921 |
| Total number of presumptive TB cases with a valid sputum smear or Xpert MTB/RIF test result | 914 | 1,831 | 2,306 | 2,154 | 7,205 |
| Total number of presumptive TB cases that were sputum smear positive (and not tested by Xpert) | 52 | 87 | 50 | 48 | 237 |
| Total number of presumptive TB cases there were Xpert MTB positive | 45 | 136 | 218 | 178 | 577 |
| Total number of presumptive TB cases that were clinically diagnosed | 64 | 169 | 235 | 228 | 696 |
| Total number of presumptive TB cases that were bacteriologically confirmed or clinically diagnosed | 161 | 392 | 503 | 452 | 1,508 |
| Total number of sputum smear positive cases that were notified and started TB treatment | 49 | 87 | 50 | 47 | 233 |
| Total number of Xpert MTP positives cases that were notified and started TB treatment | 45 | 135 | 216 | 177 | 573 |
| Total number of clinically diagnosed cases that were notified and started TB treatment | 64 | 169 | 235 | 228 | 696 |
| Total number of all bacteriologically confirmed or clinically diagnosed cases that started treatment | 158 | 391 | 501 | 452 | 1,502 |
| Total number of notified TB cases that abandoned treatment | 35 | 29 | 53 | 30 | 147 |
| Number of CHW led searches for abandoned cases | 28 | 29 | 53 | 30 | 140 |
| Number of abandoned cases that were relinked to care and restarted TB treatment | 26 | 25 | 46 | 24 | 121 |
| Number of pre-treatment lost of follow-up (PTLTFU) cases (diagnosed with TB, but not notified) | 42 | 63 | 22 | 42 | 169 |
| Number of CHW led searches for PTLTFU cases | 32 | 61 | 21 | 42 | 156 |
| Number of PTLTFU cases that were linked to care and started TB treatment | 31 | 59 | 18 | 40 | 148 |
| Number of TB index case households visited by a CHW | 166 | 283 | 375 | 299 | 1,123 |
| Number of children <5 that are household contacts of the index case that were screened for TB | 102 | 162 | 270 | 292 | 826 |
| Number of children <5 that were eligible for Isoniazid Prophylactic Therapy (IPT) | 84 | 113 | 249 | 264 | 710 |
| Number of children <5 that started IPT | 84 | 113 | 242 | 264 | 703 |
| **Calculated Percentages for Key Indicators** | **Total** | **Total** | **Total** | **Total** | **Total** |
| Percentage of presumtive TB cases with a valid smear or Xpert laboratory result | 85.7% | 86.9% | 92.5% | 95.6% | 84.4% |
| Percentage of presumptive TB cases that underwent TB testing with bacteriologically confirmed TB | 9.1% | 10.6% | 10.7% | 10.0% | 10.3% |
| Percentage of presumptive TB cases that were clinically diagnosed with TB | 6.0% | 8.0% | 9.4% | 10.1% | 8.8% |
| Percentage of presumptive TB cases that had bacteriologically confirmed or clinically diagnosed TB | 15.1% | 18.6% | 20.2% | 20.1% | 19.1% |
| Percentage of bacteriologically confirmed TB cases that were notified and started TB treatment | 96.9% | 99.6% | 99.3% | 99.1% | 99.0% |
| Percentage of clinically diagnosed TB cases that were notified and started TB treatment | 100.0% | 100.0% | 100.0% | 100.0% | 100.0% |
| Percentage of all forms TB cases that were notified and started TB treatment | 98.1% | 99.7% | 99.6% | 100.0% | 99.6% |
| Percentage of patients that abandoned treatment that were traced by CHWs and restarted TB treatment | 92.9% | 86.2% | 86.8% | 80.0% | 86.4% |
| Percentage of PTLTFU cases that were traced by CHWs and restarted TB treatment | 96.9% | 96.7% | 85.7% | 95.2% | 87.6% |
| Percentage of children eligible for IPT that started IPT | 100.0% | 100.0% | 97.2% | 100.0% | 99.0% |

Q = Quarter

increase of 656 B+ or 28.8% and 763 all forms or 14.6% respectively) in intervention districts. In contrast the number of notifications in control districts during this yearlong study fell to 806 B+ and 1877 all forms of TB (a decrease of 139 or -14.7% and 371 or -16.5% respectively). These results are summarized in Table 2.

When controlling for the notification trends for the previous three years, B+ notifications in the intervention districts were 2,933 B+ compared an expected value of 2,711 B+ meaning a moderate increase of 222 additional people with B+ TB notified or an 8.2% increase compared to the historical trend. Similarly, the intervention districts notified 5,982 individuals with all forms of TB over an expected value of 5,860, leaving a net increase of 122 individuals with TB or a 2.1% increase.

**Table 2. TB notifications in intervention and control districts during the implementation period, historical baseline and expected notifications: Estimation of additionally notified individuals with TB.**

| | Notifications | Implementation period | Historical baseline | Expected notification according to trend | Additionally notified persons with TB (unadjusted) | % of increase in notification (unadjusted) | Additionally notified persons with TB (adjusted) | % of increase in notification (adjusted) |
|---|---|---|---|---|---|---|---|---|
| Intervention districts | New Pulmonary Bac + | 2933 | 2277 | 2711 | 656 | 28.8% | 222 | 8.2% |
| | All forms of TB | 5982 | 5219 | 5860 | 763 | 14.6% | 122 | 2.1% |
| Control districts | New Pulmonary Bac + | 806 | 945 | 1203 | -139 | -14.7% | -397 | -33.0% |
| | All forms of TB | 1877 | 2248 | 2850 | -371 | -16.5% | -973 | -34.1% |

In contrast, in the control districts, given the historical trend, the expected values were 1,203 Bac+ and 2,850 individuals notified with all forms of TB, while the actual notification showed only 806 Bac+ and 1,877 all forms; 397 or -33% and 973 or -34.1% less than expected. These results are summarized in Table 2.

The additional notification / yield ratio for B+ individuals is 0.8 and 0.5 for all forms (see Table 3) meaning not all individuals with TB yielded by CHWs (806 Bac + and 1,502 all forms of TB) could be directly translated into additional TB notifications (656 Bac + and 763 All forms of TB).

More detailed graphs of the historical trends and expected vs actual notifications are included as an annex, along with the underlying data sets.

## Discussion

This practical TB case-finding implementation study used routine NTP and project data to evaluate a CHW facility-based and contact tracing TB case-finding project with intervention and control districts in Manica Province, Mozambique. The results showed a significant increase in TB notifications in the intervention districts in comparison to historical and contemporary controls. Of the individuals with TB identified by CHWs, 54% were Bac+, while nationally in Mozambique the rates are less than 40%. This was likely due to more active TB case finding and screening from CHWs, leveraging health facility based TB testing including smear and recently deployed Xpert systems. [1, 18]

The use of the additionality to yield ratio is also of interest. Often, interventions only identify the numbers of people detected without assessing whether or not the intervention had any impact on notifications, others may report additionally notified people with TB without looking at the direct yield of the intervention itself. The ratio between additionality and yield provides a sense of how much the intervention adds to routine practice in terms of notification increases. A ratio closer to one might only be seen in prisons or populations that had no access to services previously, while a ratio closer to 0 would mean that the intervention was not really

**Table 3. Project generated yield and additionally notified individuals with TB.**

| | Yield | Additionally notified persons with TB (unadjusted) | additionality/Yield |
|---|---|---|---|
| New Pulmonary Bac + | 806 | 656 | 0.8 |
| All forms of TB | 1502 | 763 | 0.5 |

identifying more people with TB, maybe just detecting them slightly earlier. The fact that the additional numbers of notified individuals with TB is higher among the B+ than among all forms of TB (0.8 vs 0.5) suggests that some "transference" of individuals with TB that would be previously classified as clinically diagnosed and counted in all forms of notified TB, are now B + given Xpert's superior sensitivity. [19] Active case finding projects will often diagnose more B+ individuals if outreach is the main intervention and people screened do not have easy access to clinicians who can diagnose them in lieu of microbiological evidence. [20, 21] As noted in Table 3, it is estimated that only a proportion of individuals with TB identified by the project (roughly 80% of the Bac+ and 50% of all forms) actually led to the overall increase in the number of people with TB notified. The remaining likely would have found their way to diagnosis and treatment through the passive notification system even in absence of the project. However, they would have been diagnosed later, and potentially at greater cost to them and in a poorer state of disease. [22, 23]

Study limitations include that both intervention and control districts were not randomly selected and may not be representative of all Mozambican health facilities. The existence of differences between intervention and control districts that could explain the results besides the exposure to the intervention cannot be completely excluded. Using historical notification data to control for trends may not necessarily project what would happen in the future. In addition, individuals with presumptive TB were not disaggregated by coming from a facility, the community, or TB contacts, limiting the ability to fully analyze these different interventions. The decrease in notified cases in control districts is not exceptional in long term notification trends and may be due to cyclical changes in health seeking behavior and/or health services response, which may have also occurred in the intervention district in the absence of the project.

As part of a large global push to increase the numbers of people with TB diagnosed, notified and treated, [24–27] our results make relevant contributions. Despite the presence of TB services at the major health facilities in these intervention districts, this project demonstrated a significant gap in the true number of people with TB who are presenting to these health facilities and those who are eventually diagnosed and notified. Table 1 showed how CHWs worked to bring back 148 people (or 88%), out of 169, that had been diagnosed but were lost before treatment initiation, a positive finding of CHW performance also documented elsewhere. [20, 21, 28–33] During the period of this intervention less than 20% of total TB notifications in the intervention districts from contact tracing and community-based active case finding, suggesting that intensified facility-based screening using CHWs explained most of the increased notification. Given our study design, it is not possible to define which intervention within the different activities CHW performed in and sometimes outside health facilities produced more of the additionally notified individuals, but it is clear that together there was an increase of people with TB diagnosed, and that they were more likely to be B+. Facility-based screening is certainly lower cost when compared to active outreach, but activities such as contact investigation remain critical to meet targets for both case finding and is essential to start TB preventive therapy. Focused active case finding, particularly using facility-based CHWs, can improve the number of people who are detected and treated for TB and can push towards the ambitious targets of the WHO's End TB Strategy and the recent United Nations High level Meeting on TB. [24]

Historically, Mozambique has relied on passive case-finding strategies for TB, and individuals self-presenting to health facilities. There have been various community TB CHW activities, but these have been fragmented, involved different strategies, and did not utilize a standardized monitoring and evaluation (M&E) framework. As recommended by the WHO and others [34–36], this has started to change with specific projects supported by USAID such as the recent Challenge TB (CTB) Project in Sofala, Nampula, Tete and Zambezia provinces. TB

notifications increased substantially during the recent CTB project which has a strong CHW community TB-case finding element, but most of the additional individuals were not B+, which is surprising given the recent scale-up of LED microscopy and Xpert MTB/RIF testing in Mozambique. [37] An international NGO called JHPIEGO has also been leading a "cough officer" project that trains health facility janitors and lay employees to do some facility-based active TB case-finding. [37] Given the results of the CTB project, the cough officer project and this project, and combined greater acknowledgement about the need for more active TB case-finding strategies in Mozambique, the NTP contracted a local Mozambican NGO to provide community TB activities in six additional provinces where CTB was not located, including Manica province with support from their recent Global Fund award. The CHW registries and M&E framework from this TB REACH project, along with instruments from CTB and JHPIEGO were adapted by a new NTP led TB CHW working group to develop the first national CHW TB guidelines in Mozambique, along with a national and standardized M&E framework for all TB CHW partners.

## Conclusion

While overall TB notifications increased in Mozambique over the past 8 years this has been primarily driven by increases in clinically diagnosed TB, and the overall trend in TB notifications is starting to plateau. This practical TB case finding implementation study using routine NTP and project data shows that a CHW facility based and contact tracing TB case-finding strategy may significantly increase TB notifications, particularly those that are bacteriologically confirmed, in central Mozambique.

## Supporting information

**S1 File.**
(XLSX)

## Acknowledgments

We would like to thank the health facilities, districts, provincial health departments, the National TB program and the Ministry of Health for their support and collaboration with this project. Most importantly we want to thank the individuals that we served, and the health care providers that participated in this project.

## Author Contributions

**Conceptualization:** B. José, I. Manhiça, C. Mutaquiha, O. Ramis, J. Cowan.

**Data curation:** J. Jones, C. Mutaquiha, M. Chidacua, J. Cowan.

**Formal analysis:** B. José, J. Jones, C. Mutaquiha, P. Zindoga, O. Ramis, J. Cowan.

**Funding acquisition:** I. Manhiça, I. Ramiro, J. Cowan.

**Investigation:** C. Mutaquiha, M. Chidacua.

**Methodology:** B. José, I. Manhiça, C. Mutaquiha, I. Eduardo, J. Creswell, Z. Z. Qin, O. Ramis, J. Cowan.

**Project administration:** B. José, I. Manhiça, J. Jones, C. Mutaquiha, I. Eduardo, J. Cowan.

**Supervision:** B. José, I. Manhiça, P. Zindoga, I. Eduardo, J. Creswell, Z. Z. Qin, I. Ramiro, M. Chidacua, J. Cowan.

**Validation:** J. Jones, P. Zindoga, J. Creswell, O. Ramis, J. Cowan.

**Writing – original draft:** C. Mutaquiha, J. Cowan.

**Writing – review & editing:** B. José, I. Manhiça, J. Jones, C. Mutaquiha, P. Zindoga, I. Eduardo, J. Creswell, Z. Z. Qin, O. Ramis, I. Ramiro, J. Cowan.

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
