## [Decision Letter · Decision Letter 0]

26 Mar 2020

PONE-D-19-34368

Using Lay Health Workers for Facility and Community Based TB Case Finding: An Evaluation in Central Mozambique

PLOS ONE

Dear Dr Cowan,

Thank you for submitting your manuscript to PLOS ONE. After careful consideration, we feel that it has merit but does not fully meet PLOS ONE’s publication criteria as it currently stands. Therefore, we invite you to submit a revised version of the manuscript that addresses the points raised during the review process.

We would appreciate receiving your revised manuscript by May 10 2020 11:59PM. To enhance the reproducibility of your results, we recommend that if applicable you deposit your laboratory protocols in protocols.io, where a protocol can be assigned its own identifier (DOI) such that it can be cited independently in the future. For instructions see: http://journals.plos.org/plosone/s/submission-guidelines#loc-laboratory-protocols

We look forward to receiving your revised manuscript.

Kind regards,

Joel Msafiri Francis, MD, MS, PhD

Academic Editor

PLOS ONE

Journal Requirements:

2. Please include in your Methods section (or in Supplementary Information files) the participating hospitals/institutions and the districts where the intervention was used. Please also describe in more detail how the intervention sessions were conducted, including how many patients participated.

"This research was supported in part by a grant from the Stop TB Partnership’s TB REACH initiative which is funded by the Global Affairs Canada, the Bill and Melinda Gates Foundation and USAID. This research was funded in part by a 2015 developmental grant from the University of Washington Center for AIDS Research (CFAR), an NIH funded program under award number AI027757 which is supported by the following NIH Institutes and Centers (NIAID, NCI, NIMH, NIDA, NICHD, NHLBI, NIA, NIGMS, NIDDK). The content is solely the responsibility of the authors and does not necessarily represent the official views of the National Institutes of Health.

Jacob Cresswell and Zhi Zhen Qin are employees of the Stop TB Partnership and oversee the TB REACH grant portfolio which supported this project. They were not involved in project implementation, but did provide general project oversight. In addition they reviewed and provided comments of this article - for this they are listed as co-authors."

4. Thank you for providing the following Funding Statement: 

"This research was supported in part by a grant from the Stop TB Partnership’s TB REACH initiative which is funded by the Global Affairs Canada, the Bill and Melinda Gates Foundation and USAID. This research was funded in part by a 2015 developmental grant from the University of Washington Center for AIDS Research (CFAR), an NIH funded program under award number AI027757 which is supported by the following NIH Institutes and Centers (NIAID, NCI, NIMH, NIDA, NICHD, NHLBI, NIA, NIGMS, NIDDK). The content is solely the responsibility of the authors and does not necessarily represent the official views of the National Institutes of Health.

Jacob Cresswell and Zhi Zhen Qin are employees of the Stop TB Partnership and oversee the TB REACH grant portfolio which supported this project. They were not involved in project implementation, but did provide general project oversight. In addition they reviewed and provided comments of this article - for this they are listed as co-authors."

We note that one or more of the authors is affiliated with the funding organization, indicating the funder may have had some role in the design, data collection, analysis or preparation of your manuscript for publication; in other words, the funder played an indirect role through the participation of the co-authors.

If the funding organization did not play a role in the study design, data collection and analysis, decision to publish, or preparation of the manuscript and only provided financial support in the form of authors' salaries and/or research materials, please review your statements relating to the author contributions, and ensure you have specifically and accurately indicated the role(s) that these authors had in your study in the Author Contributions section of the online submission form. Please make any necessary amendments directly within this section of the online submission form.  Please also update your Funding Statement to include the following statement: “The funder provided support in the form of salaries for authors [insert relevant initials], but did not have any additional role in the study design, data collection and analysis, decision to publish, or preparation of the manuscript. The specific roles of these authors are articulated in the ‘author contributions’ section.”

If the funding organization did have an additional role, please state and explain that role within your Funding Statement.

Please also provide an updated Competing Interests Statement declaring this commercial affiliation along with any other relevant declarations relating to employment, consultancy, patents, products in development, or marketed products, etc.  

6. Please amend the manuscript submission data (via Edit Submission) to include author M Chidacua.

Reviewers' comments:

Reviewer's Responses to Questions

**Comments to the Author**

1. Is the manuscript technically sound, and do the data support the conclusions?

Reviewer #1: Partly

Reviewer #2: Yes

Reviewer #3: Partly

2. Has the statistical analysis been performed appropriately and rigorously? 

Reviewer #1: No

Reviewer #2: Yes

Reviewer #3: No

3. Have the authors made all data underlying the findings in their manuscript fully available?

Reviewer #1: Yes

Reviewer #2: Yes

Reviewer #3: Yes

4. Is the manuscript presented in an intelligible fashion and written in standard English?

Reviewer #1: Yes

Reviewer #2: Yes

Reviewer #3: No

5. Review Comments to the Author

Reviewer #1: It does appear that this intervention had a substantial beneficial impact, however I have many questions. Some of these are requests for clarification, others are more substantive.

Line 54: Readers might want a brief explanation of how WHO estimates the incidence rate of TB.

Line 59: We would want to see a brief explanation of the notification system. How does this work? By implication it appears to be operated by the NTP but who are the reporters? Since it appears that not all notified cases are laboratory confirmed what are the criteria?

Line 76: Answers to the above might help us understand why there is a gap between diagnosed and notified cases.

Line 89: I would like to see a fuller description of the training the CHWs received. How are they recruited? What are their initial qualifications? How many hours of training do they undergo? Is there a manual? Are there certification requirements? How are they supervised?

Line 91: Please explain "five symptom screening."

Line 102: You should say something in the discussion about the differences between the intervention and control districts. That these were not randomly selected is a limitation of the study. You say there was no active case finding in the control districts but was there already active case finding in the intervention districts apart from the CHW intervention? What are other differences?

Line 115: Please explain the TB REACH methodology.

Line 122: What do you mean by secular notification trends? How were these extracted from the observed trends? That you refer to description of the methodology elsewhere is not sufficient, you need to at least explain the basics of what you did here. It is not clear where you actually report the difference between the observed and expected changes in notification based on the "secular" trend.

Table 2: One of the column labels is in Portuguese.

Line 173: It is surprising that the number of notifications in the control districts fell during the study period. You should at least offer some speculative explanations for this in the discussion.

Line 175: You refer to a regression analysis but you say nothing about it. What kind of regression? Again, where do the "secular trends" come from? Why are notifications in the control area an independent variable in this regression? I do not understand the reported results, that B+ notifications increased by 1.44 times, for example. This is not the usual interpretation of a regression coefficient.

Line 182. I do not understand this discussion of the "notification/yield" ratio. I thought notifications were your only outcome variable. What does the unnotified yield consist of and how is it ascertained? This also means that the discussion beginning at line 205 is indecipherable. What do you mean by additionality vs.yield? If cases are not reported how do you know they exist?

Line 229: What does JHPIEGO stand for?

Line 242: You say there is a gap between the true number of TB cases presenting to health facilities and those that are diagnosed and reported, but you do not explain how you know this. You go on to say that you do not know which intervention produced more of the additional cases but on lines 139 et seq it appears that you do disaggregate contact tracing and community based case finding from clinic-based ascertainment.

Line 266: You say that presumptive TB cases were not disaggregated as coming from a facility the community or TB contacts but again I thought you had reported these distinctions in detail starting around line 139.

In sum, I think this is likely a good contribution but you need to explain it much better.

Reviewer #2: Strength:

A well designed pragamatic study to assess an important screening intevention in facilities using CHW. The manuscripts answers an important TB research question and will contribute knowledge on how to increase case detection at facility. Studies have shown TB patients had visited facilities on several occasion before a TB diagnosis. Appropriate analysis and use of standardized evaluation framework

Weakness:

Multiple intervetions on case detection were going on or had been done in these study areas. Although the authors have discussed these in their discussion, it remains a weakness and consider revising the conclusion from ....can lead.. to ...may lead...

Specific comments

74 ‘Multiple studies have documented significant patient and health system delays in TB

75 diagnosis in Mozambique’..this sentence needs to be referenced

143 Using the TB REACH terminology, the yield of this...The terminology needs to be defined in the methods

Reviewer #3: Using Lay Health Workers for Facility and Community Based TB Case Finding: An

Evaluation in Central Mozambique

TB case finding is a priority for many countries which have lower treatment coverage. The use of community health care workers to supplement efforts by NTP is crucial to close the missing gap. CHW have the potential to improve TB case finding by conducting community active case finding especially on bacteriologically confirmed contact. The paper is important to increase the body of evidence of CHW role in TB control efforts. But there are few issues that need to be addressed to bring this paper to the required quality and better inform the TB community on the contribution of CHW on TB case finding.

GENERAL COMMENTS

• Change the language from cases to TB patients were appropriate to confirm to non-stigmatizing language.

• The paper, in some areas, misses the logical sequential flow of idea. It may difficult for some readers to follow the authors main story.

• Clearly define the main objective and exploratory objectives, as the way they are written now, they are given equal emphasis which can be a bit confusing.

SECTION SPECIFIC COMMENTS

Title: proposed to maintain community healthcare workers rather than lay health workers. The lay health workers have not referred anywhere else in the manuscript.

ABSTRACT:

Line 27-28: “Mozambique has one of the highest rates of both TB and HIV in the world 28 but an estimated TB treatment coverage of only 57% in 2018”. Needs clarity of the whether talking about incidence or mortality? Also consider rephrasing the sentence.

Line 39: The CHW screening activities yielded 1,502 notified and treated TB cases. The increase in the TB patients notified is only 763, please clarify why the different numbers?

Line 39-40: as we are looking for additionality of the intervention as compared to before the intervention, the correct number should be 763 and should be written and then the percentage.

Line 39-40: the 1,502 is this number compared to baseline, or this is the number of TB cases contributed by CHW during the intervention period. This has to be clear.

INTRODUCTION

General comment

• There is no flow as there is no connection between sentences/there is a no story line for the reader to follow what the authors want to communicate.

• I think the references are before the full stop and not after. Please review the manuscript and change accordingly.

• The article talks about the effect of CHW in increasing TB case finding, not even a single literature that highlight the effect of CHW in overall case finding is other areas with similar TB epidemiological profile.

• I seem to miss the primary objective of the

Line 52: put the abbreviation of TB here and not in line 54. It makes sense to put the abbreviation in the first use of the word in question.

Line 55: the authors are already talking about Mozambique; it is kind of a repetition to again mention Mozambique. Also, consider rewriting this sentence… “resulting in 162,000 new cases each year”. Authors can consider to rewrite the sentence into something like “…translating to an estimated 162,000 new TB patients”.

Line 56-58: “This has been largely fueled by the HIV epidemic with a national 57 prevalence of 13.2%, but compounded by deep poverty with an estimated average 58 gross national income per capita of only $440 USD”. Please rewrite this sentence. Keep it simple, and separate each part and explain it clearly. This has been largely, what exactly? Consider starting a sentence like, TB is largely fueled by high HIV epidemic of 13.2%, poverty ……” Also, the word deep poverty has not objective measurement to a reader, it is just open to too much interpretations which are not standard.

Line 58-60: “Of the estimated incident TB 59 cases, 92,381 were diagnosed and notified in 2018 resulting in a treatment coverage 60 rate of only 57%”. The sentence should go after end of Line 56 which talks about WHO estimated TB incidence rate. This is a logical flow when authors want to talk about treatment coverage.

Line 68-73: authors to consider making this the last paragraph of your introduction section.

Line 74-85: these are just too many hypotheses which seem not be related to the CHW role. I am curious to how you managed to test all these hypotheses. Also, the hypothesis lacks direction in terms whether increasing or decreasing your outcomes of interest.

METHODS

• Define the content of training package to CHW which includes the topics covered, for how many days and conducted by who. Was there any proficiency testing done to ensure the understanding of the CHW?

• The paragraphs have mixed information on study setting, population, data collection and study procedures. The authors can consider separating these into clear headings for easy of the readers to understand.

Ethics statement

Ethics statement was only waived from University of Washington IRB. What about local or national ethics bodies in Mozambique? Was there no waiver applied to these ethics bodies as the research was done there?

Study setting

Study population and study definitions

Laboratory investigation

Need to mention the laboratory procedures used to confirm for TB.

Statistical analysis

I am missing the statistical analysis of the effect of interventions within the clusters and comparison between intervention and control areas.

Line 128: the bracket is out of place.

RESULTS

General comments of results section

• Need consistency of reporting if putting numbers and proportions for each sub-population accessing a certain service.

• Intervention and control areas: consider additional tables (supplement) to describe the baseline characteristics of the intervention and control areas. You may include notification, population size and availability of diagnostic services (smear microscopy and Xpert).

• Table 1: is a bit difficult to read and make out the numbers and proportions. The table has been copied from excel and pasted here, I would suggest you have a Microsoft word table. The absolute numbers and proportions are separate, making the reading of this table difficult.

Line 137-143: consider having a flow diagram for the readers to understand the

Line 137: consider revising the text to “… CHWs identified 8532 presumptive TB patients….”

Line 138: consider adding proportion of the presumptive TB patients sent for laboratory testing

Line 140: what do you mean valid sputum smear microscopy or Xpert MTB/RIF test? So, 16% of the tests had invalid results or they had negative results?

Line 140: “… 1508 (18%) of the …” are these a sub-population of presumptive TB patients either tested for TB for of all presumptive TB patients? Consider to be specific of denominators when presenting the results of TB care cascade.

Line 143: “Using the TB REACH terminology…” is not just the additional TB patients from the intervention? Is it really a TB REACH terminology? Or is it necessary to mention that? The TB community would understand yield or additional TB patients.

Line 144: the 1,508 all forms of TB notified from the intervention group, is different from line 39 in the abstract section which is 1,502 TB notified of all forms.

Line 145: “.. and 286 contacts of TB were notified ..” I think the use of the word notified for contacts is not appropriate. I would reserve for TB patients. authors can consider using the words like reported, identified etc.

Line 178: there is mention of regression analysis which is not mentioned in the statistical analysis under methods.

Line 148-154: these are the findings from the implementation period and lacks any comparison either from the control, or the intervention period? We need some sort of baseline data to compare the effect of CHW on PTLTFU and LTFU. What were your hypothesis in terms of direction on these outcomes?

DISCUSSION

• The discussion section was expected to comment on the decreased TB notification from control districts. Why is that? What is common among the control districts that affected case finding during the project period?

• Were there any concurrent interventions that were implemented during the implementation of The TB REACH project in the intervention districts?

• Are the results consistent with other CHW efforts in other settings with similar background epidemiological profile? There are many research outputs from many settings that will better inform the performance of this Mozambique experience, but those references are missing in this paper.

• Line 266-268: reading from methods and part of the methods section, I had the impression that all presumptive TB patients were registered in the CHW registers based on the interventions done by CHW. So why the authors would not be able to fully analyze these different interventions? Authors need to clarify.

• I am not sure of the role of JHPIEGO and CTB in this project. Line 229-239 is more of lessons learnt and its impact in Mozambique in developing CHW TB guidelines. Authors need to consider to shorten this section, and if need be present this additional information in the supplement text. JHPIEGO, write in long form.

• Line 240-262: part of the discussion on the yield should go to the first two paragraphs and discuss more the results and compare with other settings.

REFERENCES

General remarks

• The name of organization or institutions should be written in full, and they are not shorted as individual names. Please check the following:

• Online reference can be made on the web references only? Please check all the references and see if they fit the journal guidelines.

Check reference no. 4. What is “Bank W”?

Check reference no. 15. What is “Partnership. ST”?

6. PLOS authors have the option to publish the peer review history of their article (what does this mean?). If published, this will include your full peer review and any attached files.

Reviewer #1: Yes: M. Barton Laws, Ph.D.

Reviewer #2: No

Reviewer #3: No

---

## [Author Response · Author response to Decision Letter 0]

27 May 2020

Dear PLOS Medicine Editorial Team,

It is our pleasure and privilege to submit this rebuttal letter for the associated manuscript “Using Community Health Workers for Facility and Community Based TB Case Finding: An Evaluation in Central Mozambique”. We are grateful for your teams detailed comments which we have responded to below and we have incorporated relevant changes in the manuscript.

Thanks for your consideration and please let us know if you have any questions or comments.

Sincerely,

James Cowan MD, MPH, MBA

Journal Requirements

Done. 

2. Please include in your Methods section (or in Supplementary Information files) the participating hospitals/institutions and the districts where the intervention was used. Please also describe in more detail how the intervention sessions were conducted, including how many patients participated.

The five intervention districts in Manica province are: Gondola, Manica, Mossurize, Bárue, and Chimoio districts. We worked in each of the district hospital/health centers and nine peripheral health centers. We have updated this language in the text.

If you would like to know the home health facility of each CHW they are the following and can be added as an annex is desired: Centro de Saude (CS or “Health Center”) Chissui, CS 7 Abril, CS Ed Mondlane, CS Nhamaonha, CS Vila Nova, CS 1 Maio, CS Dacata, Hospital Distrital Mussorize, Hospital Distrital Goi Goi, CS Mude, CS Chiurairwe, CS Penhalonga, CS Messica,CS Mavonde, CS Jecua, CS Machipanda, Hospital Distrital Manica, Hospital Distrital Manica, CS Chitunga, CS Amatongas, CS Inchope, Hospital Distrital Gondola, CS Chipindaumwe, CS Muda Seracao, CS Nhassacara, Hospital Distrital Barue, CS Nhazonia, CS Honde, CS Nhanpassa.

We provided some additional details how the intervention sessions were conducted, using WHO recommended symptom screening and contact tracing.

In the results section we explained that on average that each CHW led 19 sessions per month, and that over 277,000 individuals participated in these sensitization sessions. Of these 8,532 were identified as presumptive TB cases, with at least one positive result on symptom screening.

"This research was supported in part by a grant from the Stop TB Partnership’s TB REACH initiative which is funded by the Global Affairs Canada, the Bill and Melinda Gates Foundation and USAID. This research was funded in part by a 2015 developmental grant from the University of Washington Center for AIDS Research (CFAR), an NIH funded program under award number AI027757 which is supported by the following NIH Institutes and Centers (NIAID, NCI, NIMH, NIDA, NICHD, NHLBI, NIA, NIGMS, NIDDK). The content is solely the responsibility of the authors and does not necessarily represent the official views of the National Institutes of Health.

Jacob Cresswell and Zhi Zhen Qin are employees of the Stop TB Partnership and oversee the TB REACH grant portfolio which supported this project. They were not involved in project implementation but did provide general project oversight. In addition, they reviewed and provided comments of this article - for this they are listed as co-authors."

We provided an updated Funding Statement in the cover letter, and the same statement in the funding section of the manuscript. Please let us know if this is adequate.

4a. Thank you for providing the following Funding Statement: 

"This research was supported in part by a grant from the Stop TB Partnership’s TB REACH initiative which is funded by the Global Affairs Canada, the Bill and Melinda Gates Foundation and USAID. This research was funded in part by a 2015 developmental grant from the University of Washington Center for AIDS Research (CFAR), an NIH funded program under award number AI027757 which is supported by the following NIH Institutes and Centers (NIAID, NCI, NIMH, NIDA, NICHD, NHLBI, NIA, NIGMS, NIDDK). The content is solely the responsibility of the authors and does not necessarily represent the official views of the National Institutes of Health.

Jacob Creswell and Zhi Zhen Qin are employees of the Stop TB Partnership and oversee the TB REACH grant portfolio which supported this project. They were not involved in project implementation but did provide general project oversight. In addition, they reviewed and provided comments of this article - for this they are listed as co-authors."

We note that one or more of the authors is affiliated with the funding organization, indicating the funder may have had some role in the design, data collection, analysis or preparation of your manuscript for publication; in other words, the funder played an indirect role through the participation of the co-authors.

If the funding organization did not play a role in the study design, data collection and analysis, decision to publish, or preparation of the manuscript and only provided financial support in the form of authors' salaries and/or research materials, please review your statements relating to the author contributions, and ensure you have specifically and accurately indicated the role(s) that these authors had in your study in the Author Contributions section of the online submission form. Please make any necessary amendments directly within this section of the online submission form. Please also update your Funding Statement to include the following statement: “The funder provided support in the form of salaries for authors [insert relevant initials], but did not have any additional role in the study design, data collection and analysis, decision to publish, or preparation of the manuscript. The specific roles of these authors are articulated in the ‘author contributions’ section.”

 If the funding organization did have an additional role, please state and explain that role within your Funding Statement.

Thank you. We have updated the funding statement in the Cover Letter, the manuscript and the authors contribution section. We have clarified that Global Affairs Canada, The Bill and Melinda Gates Foundation, USAID and the NIH are the “funders”. We are not considering the Stop TB Partnership or TB REACH a “funder” but an initiative or a pass-through mechanism. The funders provided salary support for JCresswell, ZZQ and JCowan but did not have any additional role in the study design, data collection and analysis, decision to publish, or preparation of the manuscript. The specific roles of these authors are articulated in the ‘author contributions’ section.

4b. Please also provide an updated Competing Interests Statement declaring this commercial affiliation along with any other relevant declarations relating to employment, consultancy, patents, products in development, or marketed products, etc. 

No authors report any competing interests. Thus, this does not alter our adherence to PLOS ONE policies on sharing data and materials.

4c. Within your Competing Interests Statement, please confirm that this commercial affiliation does not alter your adherence to all PLOS ONE policies on sharing data and materials by including the following statement: "This does not alter our adherence to PLOS ONE policies on sharing data and materials.” (as detailed online in our guide for authors http://journals.plos.org/plosone/s/competing-interests) . If this adherence statement is not accurate and there are restrictions on sharing of data and/or materials, please state these. Please note that we cannot proceed with consideration of your article until this information has been declared.

Please see response to 4b.

4d. Please know it is PLOS ONE policy for corresponding authors to declare, on behalf of all authors, all potential competing interests for the purposes of transparency. PLOS defines a competing interest as anything that interferes with, or could reasonably be perceived as interfering with, the full and objective presentation, peer review, editorial decision-making, or publication of research or non-research articles submitted to one of the journals. Competing interests can be financial or non-financial, professional, or personal. Competing interests can arise in relationship to an organization or another person. Please follow this link to our website for more details on competing interests: http://journals.plos.org/plosone/s/competing-interests

 Noted. Please see response to 4b.

After discussing internally, we are comfortable sharing the data. In general, the TB Program Manager in Mozambique would like to be informed when any non-Mozambique Ministry of Health organization is accessing and analyzing TB data from Mozambique. If possible we would request that any group accessing this data please, as a courtesy inform Dr. Ivan Manhiça at ivanmca2004@yahoo.com.br particularly if they have plans to share or publish additional findings.

Please see response to #5.

We will upload an excel with the core data and our analysis.

 Agreed.

6. Please amend the manuscript submission data (via Edit Submission) to include author M Chidacua.

Done. 

Response to reviewers

Reviewer #1: It does appear that this intervention had a substantial beneficial impact, however I have many questions. Some of these are requests for clarification, others are more substantive.

Line 54: Readers might want a brief explanation of how WHO estimates the incidence rate of TB.

WHO produces annually national estimates of incidence for all countries which are then internationally used to calculate the gap between estimated incidence and actual notification. It declares these “estimates of TB incidence are produced through a consultative and analytical process led by WHO and are published annually [and are] based on annual case notifications, assessments of the quality and coverage of TB notification data, national surveys of the prevalence of TB disease and on information from death (vital) registration systems. Uncertainty bounds are provided in addition to best estimates. Details are available from” Policy and recommendations for how to assess the epidemiological burden of TB and the impact of TB control and Annex 1 of the WHO global tuberculosis report" . In consequence, we added a short sentence in the manuscript summarizing this explanation for readers. 

Line 59: We would want to see a brief explanation of the notification system. How does this work? By implication it appears to be operated by the NTP but who are the reporters? Since it appears that not all notified cases are laboratory confirmed what are the criteria?

Cases are diagnosed and notified in health facilities. They are either (a) clinically diagnosed because access to tests was not available at the time or because they were negative while the clinical and epidemiological criteria made the clinician consider TB as the most likely diagnosis (based on history, contact, physical exam, chest imaging etc – guidelines on how to make a clinical diagnosis are outlined in the Mozambique TB Guidelines), or (b) bacteriologically confirmed. The causes of the gap between incident cases and notified cases is multifactorial. In our article we try to show how active case finding using CHW can recover cases which would have been part of the gap. We expanded the explanation later in the section hoping to provide a clearer explanation to the reader. Notification follows the definitions and reporting framework established by WHO and we added a reference to it.

Line 76: Answers to the above might help us understand why there is a gap between diagnosed and notified cases.

See our reply to question mentioned for Line 59.

Line 89: I would like to see a fuller description of the training the CHWs received. How are they recruited? What are their initial qualifications? How many hours of training do they undergo? Is there a manual? Are there certification requirements? How are they supervised?

This project employed one main clinical supervisor who had a decade of experience providing TB care and supervisory services in Manica Province, Mozambique. This individual worked with clinicians, facility level TB nurses, and community leaders to identify potential TB CHWs in each district and in each of 14 participating health centers and hospitals (some of these candidates had worked on previous CHW project, and many were TB survivors). He then led interviews of candidates with these key partners and the district TB supervisor. 

Candidates needed to have equivalent of an 8th grade education, to be literate, have the capacity to operate a smart phone, do basic math and to be able to manage a written TB registry of presumptive TB cases. Each CHW underwent a week of training using a NTP approved training package for CHWs. The project supervisor provided a minimum of monthly in-person supervision visits, and frequently more to each CHW to review their written registries and to discuss cases. We also maintained a WhatsApp group for all CHWs and the supervisor to share best practices, for CHWs to request support for complex situations, and to disseminate updated guidance. Over the course of this project 4 CHW either left their position for personal reasons or were not regularly reporting for work and were fired after consultation with MOH district supervisors. They were all rapidly replaced, and these replacements underwent similar training.

In addition to the study supervisor, these 14 health centers also had a local TB nurse, and there is also a district TB supervisor as part of the ministry of health. On a weekly basis the CHWs also reported to these individuals, who were required to review, and sign off on the monthly summary report that the CHWs submitted to their supervisor. These reports were digitized and then served as the basis for the database analyzed by this study, along with routinely reported facility level TB data that flowed through the MOH reporting system.

Line 91: Please explain "five symptom screening."

It refers to verbal symptom screening to individuals actively asking explicitly about the presence of at least one of the following symptoms within the previous two weeks: (1) cough for more than a week; (2) weight loss, (3) night sweats, (4) fever or (5) hemoptysis. This was recommended by the WHO and we are including the reference in the revised manuscript. World Health Organization. Systematic Screening for active tuberculosis: principles and recommendations. Geneva. World Health Organization, 2013.

Line 102: You should say something in the discussion about the differences between the intervention and control districts. That these were not randomly selected is a limitation of the study. You say there was no active case finding in the control districts but was there already active case finding in the intervention districts apart from the CHW intervention? What are other differences?

Intervention and control districts could not be randomly selected and we acknowledge that is a weakness of the study and we include it in our discussion. However, control districts were selected among districts where there was no active case finding in process and were demographically and socially roughly comparable with our intervention area, including a similar level of health care deployment. There was no other active case finding activity apart from our CHW project in our intervention area. We added an explanation in the methodology and in the discussion. 

Line 115: Please explain the TB REACH methodology.

The TB REACH methodology is comprehensively reviewed in the references quoted. Essentially, it compares notification in pre and during the intervention time periods and calculates additional notifications above what could be expected if the historical notification trend would be maintained over time. It also compares the evolution of the notification trends in the evaluation population, where it is expected to grow if the intervention is effective, against the evolution in the control population, where it is expected to continue its historical trend. We expanded the explanation in the revised manuscript to make it clearer to the reader. 

Line 122: What do you mean by secular notification trends? How were these extracted from the observed trends? That you refer to description of the methodology elsewhere is not sufficient; you need to at least explain the basics of what you did here. It is not clear where you actually report the difference between the observed and expected changes in notification based on the "secular" trend.

The historical notification trend (or the secular notification trend) in the last three years both in the intervention and the control districts for the bacteriologically confirmed new cases of TB and for all forms is used to predict the expected values for the implementation period in absence of the intervention. These estimations are then compared with the true notification over the implementation period. We have updated and tried to explain this better in the manuscript.

Table 2: One of the column labels is in Portuguese.

Thank you. This is corrected.

Line 173: It is surprising that the number of notifications in the control districts fell during the study period. You should at least offer some speculative explanations for this in the discussion.

These oscillations in long term notification trends are not exceptional. In fact, a similar low can be identified in the period Q4 2015-Q2 2016 (see figures from the annex) and can be related to changes in health seeking behavior and/or overall relaxation in the mainstream health system or in the notification system. We add an explanation in the discussion. 

Line 175: You refer to a regression analysis but you say nothing about it. What kind of regression? Again, where do the "secular trends" come from? Why are notifications in the control area an independent variable in this regression? I do not understand the reported results, that B+ notifications increased by 1.44 times, for example. This is not the usual interpretation of a regression coefficient.

We updated the description of the analysis and we hope it is now clearer and simpler. We introduced a new table which we hope better summarizes the results. 

Line 182. I do not understand this discussion of the "notification/yield" ratio. I thought notifications were your only outcome variable. What does the unnotified yield consist of and how is it ascertained? This also means that the discussion beginning at line 205 is indecipherable. What do you mean by additionality vs. yield? If cases are not reported how do you know they exist?

In this project, as in any active case finding project, we cannot claim all the cases identified by the project would have not been diagnosed without the project deployment. Even in the most adverse conditions in the weakest health system, some cases are always diagnosed. In that sense, we cannot claim that all the cases identified by the project can be counted as an additionally notified case. 

We try to measure this fact with the “additional notification/yield” ratio. The numerator is the difference between the actual notified cases during the project implementation minus the notification that could be expected. The denominator is the number of TB cases identified by the active case finding project. The logic is that if all the yielded cases were truly additional notification the ratio would be 1. The result in our project, as in most projects of this type, is below 1, meaning only this proportion of the total cases identified could eventually be considered as additional. The methodology is fully explained in the references quoted in the manuscript.

We improve our explanation here and also in line 205 hoping it becomes clearer to readers and introduced this table.

Line 229: What does JHPIEGO stand for?

JHPIEGO is an international non-profit organization affiliated with Johns Hopkins University (JHU) and was initially called Johns Hopkins Program for International Education in Gynaecology and Obstetrics (JHPIEGO). Since its creation its interest grew outside its initial scope but the name remained and is used without reference to its prior meaning. We added a short insert in the text making reference to its NGO character. 

Line 242: You say there is a gap between the true number of TB cases presenting to health facilities and those that are diagnosed and reported, but you do not explain how you know this. You go on to say that you do not know which intervention produced more of the additional cases but on lines 139 et seq it appears that you do disaggregate contact tracing and community-based case finding from clinic-based ascertainment.

(a) In table 1, we could present the results of how many TB cases that had been diagnosed and were lost to follow up before treatment initiation, namely 148 out of 169 (88%). It is in that sense that we can say that the gap exists and that our project was operational in reduce it. We try to explain it better in the text.

(b) What we meant is that it was impossible to disentangle which of the CHW activities (i.e. (1)TB sensitization sessions for patients waiting at 14 health facilities, (2) active five symptom screening of people attending at all clinical sites (3) contact tracing of identified TB cases through household visits, (4) tracing PTLTFU cases and (5) tracing people that are LTFU in order to relink them to care) produced more additionally notified cases. The statement does not contradict that in the districts as a whole, according to the notification registration data, the percentage of cases identified by contact tracing and community (meaning, outside health facilities) active case finding remained similar reinforcing the view of the impact of activities CHW performed within facilities. We try to be clearer in the text. 

Line 266: You say that presumptive TB cases were not disaggregated as coming from a facility the community or TB contacts but again I thought you had reported these distinctions in detail starting around line 139.

You are right we had made a description of what CHW did in around line 139 but we could not link the impact of each one of these different activities in notification. We clarify it in the text.

In sum, I think this is likely a good contribution but you need to explain it much better.

Thank you for all your comments. We hope we have been able to use all of them to improve our manuscript.

 

Reviewer #2: Strength:

A well-designed pragmatic study to assess an important screening intervention in facilities using CHW. The manuscript answers an important TB research question and will contribute knowledge on how to increase case detection at facility. Studies have shown TB patients had visited facilities on several occasion before a TB diagnosis. Appropriate analysis and use of standardized evaluation framework

Weakness:

Multiple interventions on case detection were going on or had been done in these study areas. Although the authors have discussed these in their discussion, it remains a weakness and consider revising the conclusion from ....can lead.. to ...may lead...

We appreciate the comment and adjusted the conclusion statement.

Specific comments

74 ‘Multiple studies have documented significant patient and health system delays in TB diagnosis in Mozambique’. This sentence needs to be referenced

Yes, references are quoted now in the end of this sentence. 

143 Using the TB REACH terminology, the yield of this...The terminology needs to be defined in the methods

We hope that the changes introduced in the methodology section clarify the terminology. We have also taken out the mention to “using the TB REACH” terminology” in this particular sentence as we understand it does not add any value to the statement. 

 

Reviewer #3: Using Lay Health Workers for Facility and Community Based TB Case Finding: An

Evaluation in Central Mozambique

TB case finding is a priority for many countries which have lower treatment coverage. The use of community health care workers to supplement efforts by NTP is crucial to close the missing gap. CHW have the potential to improve TB case finding by conducting community active case finding especially on bacteriologically confirmed contact. The paper is important to increase the body of evidence of CHW role in TB control efforts. But there are few issues that need to be addressed to bring this paper to the required quality and better inform the TB community on the contribution of CHW on TB case finding.

GENERAL COMMENTS

• Change the language from cases to TB patients were appropriate to confirm to non-stigmatizing language.

We change the language to abide to non-stigmatizing language.

• The paper, in some areas, misses the logical sequential flow of idea. It may difficult for some readers to follow the authors main story.

With the changes introduced, we hope the writing follows a more logical sequential flow.

• Clearly define the main objective and exploratory objectives, as the way they are written now, they are given equal emphasis which can be a bit confusing.

We improved the section where we explain our objective.

SECTION SPECIFIC COMMENTS

Title: proposed to maintain community healthcare workers rather than lay health workers. The lay health workers have not referred anywhere else in the manuscript.

Agreed – we have updated the title.

ABSTRACT:

Line 27-28: “Mozambique has one of the highest rates of both TB and HIV in the world 28 but an estimated TB treatment coverage of only 57% in 2018”. Needs clarity of the whether talking about incidence or mortality? Also consider rephrasing the sentence.

It refers to the incidence rates. We improved the sentence where this issue is described.

Line 39: The CHW screening activities yielded 1,502 notified and treated TB cases. The increase in the TB patients notified is only 763, please clarify why the different numbers?

The difference and relationship between yield (cases identified by the CHW) and increase in notification is fully explained in the analysis section. We cannot provide a detailed explanation but have tried to succinctly explain this it in the updated abstract. 

Line 39-40: as we are looking for additionality of the intervention as compared to before the intervention, the correct number should be 763 and should be written and then the percentage.

You are right. Both the absolute number and the percentage are now in the abstract.

Line 39-40: the 1,502 is this number compared to baseline, or this is the number of TB cases contributed by CHW during the intervention period. This has to be clear.

1,502 is the number of patients identified by the CHW (yield) and it contributes to the notification increase of 763. Not all the 1,502 patients yielded will show as a notification increase because some of them would have found their way to diagnosis and treatment even in absence of the project. We hope that with the changes in the manuscript that is clarified in the methodology and analysis sections. We cannot provide the full explanation in the abstract but we do in the body of the manuscript.

INTRODUCTION

General comment

• There is no flow as there is no connection between sentences/there is a no story line for the reader to follow what the authors want to communicate.

We hope that with the changes in the manuscript that makes it easier and clearer to follow for readers.

• I think the references are before the full stop and not after. Please review the manuscript and change accordingly.

Looking at other PLOS One references it appears that the references are generally after the full stop or period. 

• The article talks about the effect of CHW in increasing TB case finding, not even a single literature that highlight the effect of CHW in overall case finding is other areas with similar TB epidemiological profile.

We refer to work in other settings with similar aims in the discussion, and have added a number of citations.

• I seem to miss the primary objective of the study

We improved the explanation of our hypothesis and aims in the Introduction section.

Line 52: put the abbreviation of TB here and not in line 54. It makes sense to put the abbreviation in the first use of the word in question.

Thank you. We followed your recommendation.

Line 55: the authors are already talking about Mozambique; it is kind of a repetition to again mention Mozambique. Also, consider rewriting this sentence… “resulting in 162,000 new cases each year”. Authors can consider to rewrite the sentence into something like “…translating to an estimated 162,000 new TB patients”.

Thank you. We agree and abide to your recommendation. 

Line 56-58: “This has been largely fueled by the HIV epidemic with a national 57 prevalence of 13.2%, but compounded by deep poverty with an estimated average 58 gross national income per capita of only $440 USD”. Please rewrite this sentence. Keep it simple, and separate each part and explain it clearly. This has been largely, what exactly? Consider starting a sentence like, TB is largely fueled by high HIV epidemic of 13.2%, poverty ……” Also, the word deep poverty has not objective measurement to a reader, it is just open to too much interpretations which are not standard.

We rewrote the sentence leaving it as free as possible of ambiguous wording.

Line 58-60: “Of the estimated incident TB 59 cases, 92,381 were diagnosed and notified in 2018 resulting in a treatment coverage 60 rate of only 57%”. The sentence should go after end of Line 56 which talks about WHO estimated TB incidence rate. This is a logical flow when authors want to talk about treatment coverage.

We changed the sentence order and it makes it easier to follow for readers.

Line 68-73: authors to consider making this the last paragraph of your introduction section.

Thank you for the suggestion. We made the change.

Line 74-85: these are just too many hypotheses which seem not be related to the CHW role. I am curious to how you managed to test all these hypotheses. Also, the hypothesis lacks direction in terms whether increasing or decreasing your outcomes of interest.

We rephrased our hypothesis and tried to justify why these were related with our aim that CHW could improve TB notification. 

METHODS

• Define the content of training package to CHW which includes the topics covered, for how many days and conducted by who. Was there any proficiency testing done to ensure the understanding of the CHW?

Please see response to reviewer #1 who had a similar question. There were a variety of in person and peer assessments to document the adequate understanding of the CHW.

• The paragraphs have mixed information on study setting, population, data collection and study procedures. The authors can consider separating these into clear headings for easy of the readers to understand.

We tried to simplify the writing of the methodology and hope is now clearer.

Ethics statement

Ethics statement was only waived from University of Washington IRB. What about local or national ethics bodies in Mozambique? Was there no waiver applied to these ethics bodies as the research was done there?

We have updated the text to note that IRB was waived by University of Washington and the Mozambican IRB. Thank you noting this discrepancy. 

Study setting

Study population and study definitions

Laboratory investigation

Need to mention the laboratory procedures used to confirm for TB.

Laboratory tests to confirm bacteriologically TB were either smear microscopy or Xpert MTB/RIF, depending on local availability. We introduced a sentence in the text explaining this protocol. 

Statistical analysis

I am missing the statistical analysis of the effect of interventions within the clusters and comparison between intervention and control areas.

We clarified the analysis description in the methodology section. We are hoping it is now clearer, including the introduction of self-explanatory result tables in the results section.

Line 128: the bracket is out of place.

(We could not identify any bracket in line 128. However, we reviewed the text and ensured all brackets were in their place and any redundant bracket was erased).

RESULTS

General comments of results section

• Need consistency of reporting if putting numbers and proportions for each sub-population accessing a certain service.

• Intervention and control areas: consider additional tables (supplement) to describe the baseline characteristics of the intervention and control areas. You may include notification, population size and availability of diagnostic services (smear microscopy and Xpert).

• Table 1: is a bit difficult to read and make out the numbers and proportions. The table has been copied from excel and pasted here, I would suggest you have a Microsoft word table. The absolute numbers and proportions are separate, making the reading of this table difficult.

We have attempted to address these issues – thank you.

Line 137-143: consider having a flow diagram for the readers to understand the table.

We improved table 1 that describes the logical flow of events. The original table is included as a sheet in the attached excel as an annex and may be easier to review than the image file in the word document.

Line 137: consider revising the text to “… CHWs identified 8532 presumptive TB patients….”

Thanks. We did it.

Line 138: consider adding proportion of the presumptive TB patients sent for laboratory testing

That was 93%. We included the proportion in the text.

Line 140: what do you mean valid sputum smear microscopy or Xpert MTB/RIF test? So, 16% of the tests had invalid results or they had negative results?

No, it means that 16% either they never provided a sample for testing (7%) or provided a sample but the result was invalid (9%) because of sample quality, test failure or other problems. We try to clarify it in the manuscript.

Line 140: “… 1508 (18%) of the …” are these a sub-population of presumptive TB patients either tested for TB for of all presumptive TB patients? Consider to be specific of denominators when presenting the results of TB care cascade.

We ensured the denominators are well described in the text.

Line 143: “Using the TB REACH terminology…” is not just the additional TB patients from the intervention? Is it really a TB REACH terminology? Or is it necessary to mention that? The TB community would understand yield or additional TB patients.

You are right “TB REACH terminology” does not add any value. We rephrase the sentence. Thank you.

Line 144: the 1,508 all forms of TB notified from the intervention group, is different from line 39 in the abstract section which is 1,502 TB notified of all forms.

We corrected it. 1,508 is diagnosed, 1502 is corrected and enrolled into treatment.

Line 145: “.. and 286 contacts of TB were notified ..” I think the use of the word notified for contacts is not appropriate. I would reserve for TB patients. authors can consider using the words like reported, identified etc.

Thank you. In addition, there was a typo. It was 826, not 286. We correct it.

Line 178: there is mention of regression analysis which is not mentioned in the statistical analysis under methods.

We rewrote the section to explain better our analysis. Hope it clarifies.

Line 148-154: these are the findings from the implementation period and lacks any comparison either from the control, or the intervention period? We need some sort of baseline data to compare the effect of CHW on PTLTFU and LTFU. What were your hypothesis in terms of direction on these outcomes?

We introduced an improved results description with 2 tables allowing comparisons between control and intervention districts and between historical baseline and implementation. Unfortunately, we could not have a baseline to compare the case recovery of PTLTFUs and LTFUs in this area – this data is not routinely collected by the NTP. 

DISCUSSION

• The discussion section was expected to comment on the decreased TB notification from control districts. Why is that? What is common among the control districts that affected case finding during the project period?

We have introduced our explanation of the reductions in the comparison districts. 

• Were there any concurrent interventions that were implemented during the implementation of The TB REACH project in the intervention districts?

No. Our was the only active case finding intervention in the area. We clarified it in the methodology section.

• Are the results consistent with other CHW efforts in other settings with similar background epidemiological profile? There are many research outputs from many settings that will better inform the performance of this Mozambique experience, but those references are missing in this paper.

We make some references to similar interventions in other settings, see reference in the discussion section. These results are similar to CHW case-finding efforts in other settings but these are challenging to compare given the different local contexts, scale, and differences in the interventions.

• Line 266-268: reading from methods and part of the methods section, I had the impression that all presumptive TB patients were registered in the CHW registers based on the interventions done by CHW. So why the authors would not be able to fully analyze these different interventions? Authors need to clarify.

That is now clarified in the text. CHW registers only recorded presumptive patients in active case finding within the facilities and their cascade of care till diagnose and treatment. That is analyzed and presented. For the other activities, contact tracing was reported but not how many contacts were eventually considered active TB patients, although these were reported as such in the official system. As far as LTFU recoveries their activities prevented losses to follow up, as shown in Table 1, but were not directly contributing to newly notified cases.

 • I am not sure of the role of JHPIEGO and CTB in this project. Line 229-239 is more of lessons learnt and its impact in Mozambique in developing CHW TB guidelines. Authors need to consider to shorten this section, and if need be present this additional information in the supplement text. 

We included this description to put our work in perspective of the changes taking place in Mozambique. We tried, however, to shorten the section.

JHPIEGO, write in long form.

JHPIEGO now claims to be named by this name instead of the old “Johns Hopkins Program for International Education in Gynaecology and Obstetrics”. We introduced, however, a note making clear it is an international NGO.

• Line 240-262: part of the discussion on the yield should go to the first two paragraphs and discuss more the results and compare with other settings.

You are right. That seems more logical and we changed the order.

REFERENCES

General remarks

• The name of organization or institutions should be written in full, and they are not shorted as individual names. Please check the following:

• Online reference can be made on the web references only? Please check all the references and see if they fit the journal guidelines.

We checked and corrected where appropriate. Thanks.

Check reference no. 4. What is “Bank W”?

That was supposed to mean The World Bank. We corrected it.

Check reference no. 15. What is “Partnership. ST”?

 That was supposed to mean Stop TB Partnership. We corrected it.

---

## [Decision Letter · Decision Letter 1]

6 Jul 2020

Using community health workers for facility and community based TB case finding: An evaluation in central Mozambique

PONE-D-19-34368R1

Dear Dr. Cowan,

We’re pleased to inform you that your manuscript has been judged scientifically suitable for publication and will be formally accepted for publication once it meets all outstanding technical requirements.

Kind regards,

Joel Msafiri Francis, MD, MS, PhD

Academic Editor

PLOS ONE

Additional Editor Comments (optional):

Thank you for all the revisions.

Reviewers' comments:

Reviewer's Responses to Questions

**Comments to the Author**

1. If the authors have adequately addressed your comments raised in a previous round of review and you feel that this manuscript is now acceptable for publication, you may indicate that here to bypass the “Comments to the Author” section, enter your conflict of interest statement in the “Confidential to Editor” section, and submit your "Accept" recommendation.

Reviewer #1: (No Response)

Reviewer #2: (No Response)

Reviewer #3: All comments have been addressed

2. Is the manuscript technically sound, and do the data support the conclusions?

Reviewer #1: Yes

Reviewer #2: Yes

Reviewer #3: Yes

3. Has the statistical analysis been performed appropriately and rigorously? 

Reviewer #1: Yes

Reviewer #2: Yes

Reviewer #3: Yes

4. Have the authors made all data underlying the findings in their manuscript fully available?

Reviewer #1: Yes

Reviewer #2: Yes

Reviewer #3: Yes

5. Is the manuscript presented in an intelligible fashion and written in standard English?

Reviewer #1: Yes

Reviewer #2: Yes

Reviewer #3: Yes

6. Review Comments to the Author

Reviewer #1: Thank you for the revisions. This is now much clearer and presents a more complete description of the project. Note there is a typo on line 178, and you are inconsistent in the use of B+ and Bac+.

Reviewer #2: (No Response)

Reviewer #3: The authors have revised manuscript has addressed all the comments with additional analysis and tables. No additional comments from me.

7. PLOS authors have the option to publish the peer review history of their article (what does this mean?). If published, this will include your full peer review and any attached files.

Reviewer #1: **Yes: **M.Barton Laws, Ph.D.

Reviewer #2: No

Reviewer #3: No

---

## [Editor Report · Acceptance letter]

8 Jul 2020

PONE-D-19-34368R1 

Using community health workers for facility and community based TB case finding: An evaluation in central Mozambique 

Dear Dr. Cowan:

I'm pleased to inform you that your manuscript has been deemed suitable for publication in PLOS ONE. Congratulations! Your manuscript is now with our production department. 

Kind regards, 

on behalf of

Dr. Joel Msafiri Francis 

Academic Editor

PLOS ONE